# Impact of the COVID-19 Pandemic on Self-Confidence in Patient Treatment in the Endodontic and Restorative Procedures of Dentistry Students at the University of Novi Sad

**DOI:** 10.3390/jcm13144030

**Published:** 2024-07-10

**Authors:** Milica Jeremic Knezevic, Milana Bojinovic, Daniela Djurovic Koprivica, Aleksandra Maletin, Aleksandar Knezevic, Marija Sarac, Tatjana Puskar, Zeljka Nikolasevic, Bojana Ramic

**Affiliations:** 1Faculty of Medicine, University of Novi Sad, 21000 Novi Sad, Serbia; bojinovicmilana3@gmail.com (M.B.); daniela.djurovic-koprivica@mf.uns.ac.rs (D.D.K.); aleksandra.maletin@mf.uns.ac.rs (A.M.); aleksandar.knezevic@mf.uns.ac.rs (A.K.); marija.sarac98@gmail.com (M.S.); tatjana.puskar@mf.uns.ac.rs (T.P.); zeljka.nikolasevic@mf.uns.ac.rs (Z.N.); bojana.ramic@mf.uns.ac.rs (B.R.); 2Medical Rehabilitation Clinic, University Clinical Center of Vojvodina, 21000 Novi Sad, Serbia; 3Dentistry Clinic of Vojvodina, 21000 Novi Sad, Serbia

**Keywords:** COVID-19 pandemic, self-assessment, self-confidence, dental students

## Abstract

(1) **Background**: The COVID-19 pandemic has brought about a change in the concept of teaching with the introduction of online lectures and “distance learning.” The aim of this study was to determine the impact of the COVID-19 pandemic on the confidence and both theoretical and practical knowledge of dental students regarding the courses in conservative dentistry and endodontics. (2) **Methods**: The study was conducted using an originally created online questionnaire consisting of 30 questions that the students used to make a subjective assessment of their confidence in performing both endodontic and conservative procedures using numerical scales. They were divided into two groups, a non-COVID group of students, who attended classes before the outbreak of the pandemic, and a COVID group, whose classes were suspended due to the lockdown. The data were processed in the SPSS statistic 25 program, with statistical significance *p* < 0.05. (3) **Results**: The comparison of the self-assessment of 96 students from the COVID group and 91 students from the non-COVID group revealed significant results. Compared with the COVID group, the non-COVID students felt more confident in the application of anesthesia in both the maxilla and mandible, in the initial treatment of patients in pain, as well as in all the steps of the endodontic treatment. Both groups felt equally confident in diagnostic and conservative procedures, and the level of theoretical knowledge acquired in the courses. (4) **Conclusions**: Changes in teaching due to the COVID-19 pandemic had an unfavorable effect on students’ self-confidence in performing routine dental interventions, especially in the endodontic field. The lack of confidence in the COVID group regarding the anesthetic and endodontic procedures appears due to their inability to do any of these procedures during the COVID period; we organized summer practical school under the supervision of teachers, and they gained the necessary knowledge and self-confidence for these procedures.

## 1. Introduction

On 11 March 2020, the World Health Organization declared the COVID-19 pandemic. COVID-19 is defined as an infectious disease caused by the SARS-CoV-2 virus (severe acute respiratory syndrome coronavirus 2). The virus is transmitted through droplets, contact routes, and aerosols. This disease manifests in various forms, ranging from mild to moderate, and in some cases, a severe form with fatal consequences. Approximately 80% of the patients experience a mild clinical presentation, not necessitating hospitalization, while 15% require hospital treatment and oxygen therapy. A further 5% of the patients necessitate intensive care unit treatment with respiratory support and mechanical ventilation [1,2].

Following the declaration of the pandemic, the Republic of Serbia implemented various infection prevention measures, including enhanced hygiene practices, disinfection protocols, and restrictions on movement and indoor gatherings. Consequently, the Faculty of Medicine in Novi Sad, along with other educational institutions, temporarily halted in-person classes across all the study programs. This shift necessitated a change in teaching methods, with the adoption of online lectures and distance learning modalities [3]. The shift to online teaching posed a significant challenge for the dental education program, given its reliance on hands-on work with patients, which carries a high risk of infection. Despite the Faculty of Medicine’s considerable efforts in delivering high-quality online classes, featuring case presentations and frequent student discussions, this did not fully compensate for the absence of clinical practice, essential for the development of skills and competencies required for future dentists. From 2020 to 2022, the prevalence and severity of the pandemic fluctuated, allowing for occasional practical sessions, especially for final-year students, during periods of relative calm. However, these sessions were conducted under stringent preventive measures and at a reduced capacity. In addition to acquiring theoretical and practical skills in patient care, it is essential for students completing dental studies to develop a certain level of self-confidence. The degree of confidence in clinical practice directly correlates with individual personality traits and the amount of clinical experience gained during their education. The COVID-19 pandemic significantly impacted both the quantity and quality of clinical experience for students. Consequently, individuals with particular personality profiles may graduate from university with a notably low level of confidence in their clinical abilities. Considering the global nature of the pandemic and its impact, this issue represents a challenge of global proportions [4].

Integrated academic studies in Dental Medicine at the Faculty of Medicine, University of Novi Sad, encompass the acquisition of both theoretical and clinical practical knowledge and skills. Over the course of 10 semesters, practical sessions are incrementally introduced, progressing from preclinical to clinical settings, transitioning from working with phantoms and models to engaging with real patients. This educational process takes place at the Dentistry Clinic of Vojvodina, renowned for its highly specialized specialist consultation and inpatient care services at the tertiary level of dental healthcare. Additionally, the clinic is actively involved in educational and scientific research endeavors within the field of dentistry.

Self-efficacy refers to an individual’s perception of their capability to plan and execute actions necessary to accomplish specific tasks. Academic self-efficacy pertains to evaluating one’s own abilities within academic spheres. It is generally believed that a strong sense of academic self-efficacy positively influences academic achievement, and conversely, high academic achievement can reinforce one’s sense of academic self-efficacy [5]. This study focuses on students’ capacity to assess their own work against the established standards and criteria, namely, their ability to engage in the self-assessment of acquired knowledge and skills. The insights gained from this study could prove invaluable in designing and refining curriculum structures, particularly during times of exceptional circumstances, such as the COVID-19 pandemic.

The aim of the research was to assess the influence of the COVID-19 pandemic on students’ confidence levels in endodontic and conservative procedures, as well as the theoretical and practical knowledge of the students who participated in classes during the pandemic.

## 2. Materials and Methods

The research was carried out as a cross-sectional study in the period from October to December 2022, and included 187 dental medicine students (female 76.1%).

The inclusion criterion was enrollment in the final semester of Integrated Dental Studies at the Faculty of Medicine, University of Novi Sad, during the years 2018, 2019, 2021, or 2022. The participants were recruited through personal contacts and informal social network groups/profiles. There were no age restrictions, and the respondents’ anonymity was ensured.

The Commission for the Ethics of Clinical Trials of the Faculty of Medicine of the University of Novi Sad has approved this research (number: 01-39/292/1) and all the subjects signed informed consent to participate in this research.

The subjects were divided into two groups. The first group (COVID group) consisted of the students from the generations in which the COVID-19 pandemic was declared during the sixth or eighth semester (the students who attended the last semester in 2022, respectively, instead of 2021 when classes were suspended and conducted online, which interrupted the contact of students with patients in the context of clinical exercises). The second group (non-COVID group) consisted of the students whose classes were not interrupted due to the COVID-19 pandemic, i.e., the students who attended the last semester in 2019 or 2018 year. The students who attended the last semester of 2020 were not included in the study, as the COVID-19 pandemic was declared on 15 March 2020, in the last semester, so the students could not be placed in any group.

The total number of respondents who met the given criterion was 187—96 respondents belonged to COVID, and 91 to the non-COVID group.

The data were analyzed using the commercial statistical software SPSS Statistics 25 (IBM SPSS—Statistical Package for Social Sciences, Chicago, IL, USA).

### Questionnaire on Self-Assessment of Competence for Independent Patient Care after Completion of Dental Studies

The questionnaire showed good internal consistency, with a Cronbach’s Alpha score of α = 0.83.

All the subjects fulfilled the questionnaire designed by Ilic et al. [6] which was modified for the purpose of this research. It comprises 30 questions divided into five groups. The first group (3 questions) focused on gathering data regarding the number of definite completions and fully executed endodontic therapies during the course of the study. The second group (6 questions) assessed self-perceived abilities in taking patient histories, diagnosing conditions, formulating appropriate treatment plans, providing initial pain relief to patients, and ensuring safety in prescribing necessary antibiotic therapy when indicated. The third group of questions (7 questions) pertained to the self-assessment of clinical proficiency in fundamental conservative dental procedures, including administering anesthesia in the upper and lower jaw, handling placement instruments during cavity preparation, distinguishing between healthy and caries-affected dental tissues, selecting appropriate materials for definitive fillings, and placing a metal matrix during Class II filling preparation. Additionally, the fourth group of questions (8 questions) focused on the self-assessment of proficiency in each stage of endodontic therapy, such as preparing adequate access cavities relative to tooth groups, determining working length, preparing and irrigating root canals, and properly obturating root canals with guttapercha points. The respondents rated the questions from the second, third, and fourth groups on a scale of 1 to 3, where 1 indicated disagreement, 2 represented uncertainty, and 3 indicated full agreement. Lastly, the fifth group of questions (6 questions) explored the respondents’ impressions during patient interactions and in the event of complications during conservative or endodontic procedures. The responses to these questions were rated on a scale of 1 to 5, with 1 indicating complete disagreement, 2 indicating partial disagreement, 3 reflecting uncertainty, 4 representing partial agreement, and 5 indicating complete agreement.

## 3. Results

The data were analyzed using the commercial statistical software SPSS Statistics 25. Descriptive statistics were employed to identify statistically significant differences between the groups using the T-test, with a significance level set at *p* < 0.05.

Out of the total 187 surveyed students who completed their final semester of Dental Studies, 96 (51.3%) belonged to the COVID group, while 91 (48.9%) were classified in the non-COVID group. The sample consisted of 142 women and 45 men; however, this criterion was not considered in the analysis, as the primary focus of this study was to compare the differences between the students who attended classes before and during the COVID-19 pandemic. The primary criterion for grouping the respondents was their attendance during the COVID-19 pandemic.

The descriptive analysis illustrates the disparity between the student groups concerning the quantity of dental fillings placed throughout their studies. Among the students whose classes were interrupted by the COVID-19 pandemic, the majority completed between 11 and 20 dental fillings during their exercises at the faculty (47.9%). Conversely, the students from the non-COVID group predominantly performed more than 20 dental fillings (95.6%).

The descriptive analysis highlights the contrast between the student groups concerning the frequency of endodontic treatments performed throughout their studies. Among the students whose classes were disrupted by the COVID-19 pandemic, a notable proportion (58.3%) did not have the opportunity to perform any endodontic treatments during the study period. Conversely, the students from the non-COVID group predominantly filled more than 10 root canals (47.3%).

The analysis of differences in the self-assessment abilities between the COVID and non-COVID students regarding taking adequate medical and dental histories, diagnosing dental diseases, and determining appropriate therapy plans for patients did not yield statistical significance.

In terms of providing appropriate first aid to patients experiencing dental pain, the analysis revealed no statistically significant difference between the COVID and non-COVID groups. The non-COVID group rated themselves as feeling more confident and willing to administer first aid in such situations. However, there was no statistically significant difference between the groups regarding the safety of prescribing antibiotic therapy (Table 1).

A significant difference was observed in the safety levels of administering plexus and mandibular anesthesia between the non-COVID and COVID groups (Table 2).

Upon analyzing the self-assessed proficiency in performing conservative procedures (instrument control, differentiation between caries-affected and healthy dental tissues, material selection for definitive fillings, and the placement of a metal matrix), no statistically significant differences were identified between the COVID and non-COVID student groups.

The non-COVID group rated their proficiency and competence higher than the COVID group across all the stages of endodontic therapy, including the preparation of the access cavity, determination of working length and master instrument size, proper medication and irrigation of the root canal, and hermetic obturation (Table 3).

In terms of the self-assessment of theoretical knowledge acquired at the faculty, no statistically significant difference was observed between the COVID and non-COVID groups.

The survey results indicated that the COVID group exhibited statistically significantly lower self-confidence overall when conducting endodontic therapy. However, when evaluating self-confidence in performing conservative procedures, as well as the feelings of anxiety when complications arise during work, no statistically significant difference was observed between the groups under examination (Table 4).

## 4. Discussion

After completing 10 semesters of integrated academic studies of dentistry in Novi Sad, it is necessary that students, in addition to extensive theoretical knowledge, acquire certain manual skills to become adequately trained for the quality and professional performance of general dental interventions. It is the practical teaching during education that should provide students with certain clinical experience, and the main prerequisite for this is that during the implementation of exercises at the faculty, students practice direct work on patients under the professional supervision of the Teaching Staff [7]. With the outbreak of the COVID-19 pandemic, there were some changes in teaching at the Faculty of Medicine in Novi Sad. Online classes only allow theoretical learning through lectures, discussions, and case presentations, but without direct contact with patients. During the pandemic, in addition to classical online classes, to educate dental students, numerous digital technologies were applied around the world in the form of mobile applications, virtual clinics, and patients [8]. However, none of the aforementioned modern and, above all, virtual approaches to the patient have succeeded in replacing direct work on them [9].

This research examined the potential negative impact of the pandemic on the entire education of dental students in terms of acquiring knowledge, skills, readiness, and safety for the future job of a dental doctor. Consequently, two groups of students were examined through the originally created online questionnaire, and their self-assessment of skills during work in the areas of conservative dentistry and endodontics was compared.

Speaking of the number of dental fillings placed during the course of the study, it is understandable that the non-sighted group of students placed more fillings. The reason for this lies in the continuous implementation of practical classes during their studies, as well as in the fact that they were conditioned to be able to access the exam if they met the prescribed norm of the number of clinical papers (40 completions). In contrast, in the COVID generations, clinical teaching has been discontinuous, and these norms in line with the changed situation have been drastically reduced. However, although there is a difference in the number of set completions between the observed groups of students, their self-assessment of the level of confidence when performing conservative dental procedures was the same in this study, i.e., both groups felt that they had a sufficient level of confidence for independent work in the field of conservative dentistry after graduation. The reason for this result can be the maintenance of preclinical exercises on phantoms, as well as certainly a high probability that the patient who reports for an examination has at least one carious tooth, even in times of pandemic and the periodic maintenance of practical classes.

Looking at the complexity of each step during endodontic therapy, including reduced visibility, delicate and often challenging access to workspaces, as well as the necessity for multi-session treatment, it is evident that practical work during exercises at the faculty is pivotal in achieving a certain level of safety for work [10]. Due to the situation with the COVID-19 pandemic, this paper reveals a discrepancy between the groups of students in terms of the number of processed and filled root canals. While the non-COVID generations of students in this area also had a prescribed norm that they needed to meet for taking exams in the subjects of Endodontics II (10 filled root canals), the COVID students had this norm abolished, with the highest percentage (58.3%) of students from this group not processing or filling any root canals during the course of their studies. This was reflected in the survey results, which indicated that performing all the steps in the implementation of endodontic therapy did not pose a difficulty for the students in the non-COVID group. On the other hand, the survey results showed that the COVID group of students exhibited significant uncertainty and a very low level of confidence when it came to all the endodontic procedures. The skills and confidence in endodontic therapy gained by the non-COVID group of students were acquired through constant work and/or observation in clinical exercises. Given that interaction with patients was limited during the pandemic, most students from the COVID group remained deprived of work in this area, resulting in a significant decrease in the safety of this group of students in the field of endodontics. The fact that the lack of confidence in the COVID group regarding the anesthetic and endodontic procedures appears due to their inability to do any of these procedures during the COVID period, we organized summer practical school under the supervison of teachers, and they achieved the necessary knowledge and self-confidence for these procedures.

In terms of taking adequate medical and dental histories, making diagnoses of dental diseases, and determining appropriate therapy plans for given patients, both groups of students have demonstrated a high level of confidence, indicating that they believe they possess sufficient knowledge to perform these procedures. This can be attributed to the emphasis placed on acquiring history-taking skills from the early years of study, particularly since the second year, through interactions with professional staff at the faculty. Additionally, making diagnoses and devising therapy plans relies heavily on theoretical knowledge, which has been consistently provided to all the generations of students thanks to the dedication of the professional staff at the Faculty of Medicine, University of Novi Sad.

Regarding the students’ self-assessment of their ability to provide first aid for patients experiencing tooth pain, the results indicate a higher level of confidence in the non-COVID group of students. This can be attributed to the continuous practical classes they attended, where they had more frequent exposure to such circumstances, allowing them to learn necessary procedures and overcome any apprehension related to treating these patients, who are often distressed and challenging to cooperate with. As a result, they graduated from the faculty feeling more prepared and motivated to assist patients in these cases compared to the COVID group of students.

The application of local anesthetic is often an unavoidable procedure in dentistry. In the upper jaw, this is usually performed through plexus anesthesia above the tooth undergoing intervention. Conversely, administering anesthetic in the lower jaw necessitates regional administration near the inferior alveolar nerve. This procedure can often be painful and unpleasant, and students frequently fear the possibility of inadvertently injecting anesthetic into a blood vessel, potentially leading to systemic adverse reactions affecting the cardiovascular, gastrointestinal, autonomic, and central nervous systems [11]. Our results indicate that the non-COVID students assessed themselves as significantly more capable than the COVID group in independently administering anesthetics in both jaws. Given their extensive experience with completed fillings, root canals, and encounters with painful conditions, this result is expected.

The findings of this research suggest that the students from both groups felt they had acquired sufficient theoretical knowledge, indicating that online teaching is an effective method for imparting theoretical knowledge, with its quality being comparable to that of traditional lectures. Similar conclusions were drawn by Schlenz et al. [12] in line with the findings of their study.

One of the shortcomings of our study, as well as similar ones, is the reliance solely on the self-assessment of knowledge and self-confidence. This implies that subjective attitudes are included in the analysis rather than the objectively assessed competence of students [13].

Also, as a limitation of the study, we should consider the fact that the students who participated in the study could show a tendency towards a better self-assessment of knowledge and skills because they are at the final age of their studies, which in their opinion could potentially affect the success and speed of completing their studies. Accordingly, this particular limitation, which concerns the assessment of their original competence/confidence, etc., at the time of graduation, will have intrinsic biases that cannot be ignored and is highlighted in the framework of potential limitations.

Comparing our study with others on this topic reveals certain similarities. Ilić J. et al. [6], in their research at the Faculty of Dentistry in Belgrade, also surveyed students whose classes were disrupted due to the pandemic and obtained similar results regarding the level of self-confidence of these students in clinical work in the field of endodontics. Additionally, their study did not assess the impact of the pandemic on history-taking, making adequate diagnoses, and treatment planning. Similar results were found in other regions as well. Pandarathodiyil, A.K et al. [14], in their study on the Malaysian student population, reported that the majority of students considered themselves competent for independent conservative tooth restoration. However, like in our study, the subjects perceived the lowest level of competence in conducting endodontic therapy, especially on multi-rooted teeth.

Generally, the online education of dental students during the pandemic has contributed to achieving a satisfactory level of theoretical knowledge. However, as established in the results of other studies, the curriculum aimed at acquiring practical clinical skills was impossible to achieve without direct contact with patients. Also, an interesting type of education according to Nikolopoulou is hybrid or blended education and it has become an important mode of teaching and learning during and after the COVID-19 pandemic, especially since this approach has rapidly spread in tertiary education [15].

## 5. Conclusions

The COVID-19 pandemic, which resulted in class suspensions and a shift to online learning, has led to a decrease in students’ confidence when performing certain routine dental procedures. This is particularly noticeable in the administration of anesthesia in the upper and lower jaw, endodontic procedures, and providing first aid to patients experiencing pain. However, the pandemic did not impact the acquisition and assimilation of theoretical knowledge in the field of dentistry.

Therefore, we conclude that despite the professors’ efforts to substitute clinical exercises with online methods as an effective means of imparting theoretical knowledge, dental students still lack a certain amount of clinical experience, affecting their sense of confidence in their work.

The pandemic has negatively affected students’ self-confidence in the field of endodontics, but it has not impacted their confidence in performing conservative procedures, nor has it affected the level of theoretical knowledge acquired during their studies.

## Figures and Tables

**Table 1 jcm-13-04030-t001:** Discrepancies between COVID and non-COVID groups regarding proficiency in first aid and antibiotic therapy.

Group		Mean	SD	Median	*t*-Test (df)	*p*
First aid	COVID	1.76	0.66	2.00	−3.211 (185)	0.002
	Non-COVID	2.08	0.69	2.00		
Antibiotic therapy	COVID	1.90	0.66	2.00	−0.587 (185)	0.558
	Non-COVID	1.96	0.74	2.00		

**Table 2 jcm-13-04030-t002:** Comparison of the COVID and non-COVID groups in administering plexus and mandibular anesthesia.

Group		Mean	SD	Median	*t*-Test (df)	*p*
Plexus anesthesia	COVID	2.42	0.75	3.00	−4.947 (145.07)	0.000
	Non-COVID	2.85	0.39	3.00		
Mandibular anesthesia	COVID	1.45	0.68	1.00	−3.865 (185)	0.000
	Non-COVID	1.87	0.81	2.00		

**Table 3 jcm-13-04030-t003:** Discrepancy between the COVID and non-COVID groups regarding their proficiency in endodontic procedures.

Group		Mean	SD	Median	*t*-Test (df)	*p*
Access cavities for incisors, canines, and premolars	COVID	2.00	0.77	2.00	−5.094 (185)	0.000
	Non-COVID	2.53	0.64	3.00		
Access cavities for molars	COVID	1.82	0.73	2.00	−2.424 (185)	0.016
	Non-COVID	2.09	0.77	2.00		
The determination of working lengths	COVID	1.86	0.72	2.00	−6.631 (185)	0.000
	Non-COVID	2.49	0.57	3.00		
Determining master instrument	COVID	1.88	0.73	2.00	−5.566 (185)	0.000
	Non-COVID	2.44	0.65	3.00		
The choice of solution for the irrigation and medication of root canal	COVID	1.74	0.74	2.00	−6.549 (185)	0.000
	Non-COVID	2.42	0.67	3.00		
Adequate root channel obturation	COVID	1.69	0.76	2.00	−6.630 (177.38)	0.000
	Non-COVID	2.34	0.58	2.00		

**Table 4 jcm-13-04030-t004:** Comparison of the COVID and non-COVID groups in the students’ confidence levels for performing conservative, endodontic procedures, and their feelings of distress.

Group		Mean	SD	Median	*t*-Test (df)	*p*
Confidence in performing conservative procedures premolars	COVID	3.75	1.11	4.00	−0.273 (185)	0.785
	Non-COVID	3.79	0.94	4.00		
Confidence in performing endodontic procedures	COVID	2.02	1.12	2.00	−5.865 (185)	0.000
	Non-COVID	2.96	1.05	3.00		
The feeling of distress when complications arise	COVID	3.63	1.12	4.00	1.473 (177.38))	0.143
	Non-COVID	3.37	1.22	3.00		

## Data Availability

The data that support the findings of this study are available from the corresponding author upon reasonable request.

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
