# Peer review of "Impact of the COVID-19 Pandemic on Self-Confidence in Patient Treatment in the Endodontic and Restorative Procedures of Dentistry Students at the University of Novi Sad"

_jcm, 2024, doi:10.3390/jcm13144030_

Round 1

Reviewer 1 Report

Comments and Suggestions for Authors

This paper is generally very well done, but some minor changes are in order relative to the formatting and presentation of the tabular data and the final conclusions.

While the English language proficiency is fine, the paper is laced with words inappropriately hyphenated. This appears to be due to reformatting the text from one computer program to another or one pattern of spacing to another. Examples of inappropriate hyphenation can be found on lines 35, 37 and 39. Similar errors appear throught the paper.

The tables need to be re-formatted as follows:

Titles contain too much narrative content. The titles should be shorter and more specific, with the remaining content provided in the narrative following the table. For example, the Title of Table 1 should be shortened to "Discrepancies Between Covid and non-Covid Groups Regarding Proficiency in First Aid and Antibiotic Therapy.

The table:  1) "F, T, df and P" should be spelled out or noted in a footnote to the table.  2) Actual numbers comparing Covid to non-Covid groups should be provided so readers can better understand the differences between the two groups.

The fact that the lack of confidence in the Covid group relative to the anesthetic and endodonic procedures appears due to their inability to do any of these procedures under supervision during the Covid period. This finding suggests that, for such students, some way must be found for these procedures to be done under supervision prior to granting graduation and full license to practice dentistry. This should be expanded in the discussion and highlighted in the Abstract. 

This is a very important paper with likely interest relative to the education and certification of other categories of health professionals, provided that my comments relative to the importance of clinical practice under supervision prior to graduation and license to practice are emphasized in the Discussion and Abstract sections of the paper.

This is a very important paper with likely interest relative to the education and certification of other categories of health professionals, provided that my comments relative to the importance of clinical practice under supervision prior to graduation and license to practice are emphasized in the Discussion and Abstract sections of the paper.

Author Response

  1. This paper is generally very well done, but some minor changes are in order relative to the formatting and presentation of the tabular data and the final conclusions.

  • Dear Editor and reviewers,

Thank you for the revision of our submitted manuscript. The authors feel that all the remarks made by reviewers are well-intended and valuable. We did our best to clarify the issues raised by the reviewers and to improve the quality of our manuscript.

  1. While the English language proficiency is fine, the paper is laced with words inappropriately hyphenated. This appears to be due to reformatting the text from one computer program to another or one pattern of spacing to another. Examples of inappropriate hyphenation can be found on lines 35, 37 and 39. Similar errors appear throught the paper.

- Thank you for this observation, we corrected in the text.

  •  
  1. The tables need to be reformatted as follows:

Titles contain too much narrative content. The titles should be shorter and more specific, with the remaining content provided in the narrative following the table. For example, the Title of Table 1 should be shortened to "Discrepancies Between Covid and non-Covid Groups Regarding Proficiency in First Aid and Antibiotic Therapy.

- Thank you for this observation, we corrected in the tables.

The table:  1) "F, T, df and P" should be spelled out or noted in a footnote to the table. 

2) Actual numbers comparing Covid to non-Covid groups should be provided so readers can better understand the differences between the two groups.

- Thank you for this observation, we corrected it in the article.

  1. The fact that the lack of confidence in the Covid group relative to the anesthetic and endodonic procedures appears due to their inability to do any of these procedures under supervision during the Covid period. This finding suggests that, for such students, some way must be found for these procedures to be done under supervision prior to granting graduation and full license to practice dentistry. This should be expanded in the discussion and highlighted in the Abstract.
  • - Thank you for this observation. We added explanation in the discussion and in abstract.

  1. This is a very important paper with likely interest relative to the education and certification of other categories of health professionals, provided that my comments relative to the importance of clinical practice under supervision prior to graduation and license to practice are emphasized in the Discussion and Abstract sections of the paper.

- Thank you for this observation, we agree with Your comments.

Reviewer 2 Report

Comments and Suggestions for Authors

The authors have conducted an interesting study evaluating the impact of the COVID-19 pandemic on dental education, specifically in the domains of endodontics and restorative dentistry. This is indeed an important topic in medical and dental education.

I have several concerns which require clarification:

1. The details regarding the coursework in the University should come in later after the context and topic have been explained. The introduction should be restructure so that the context is clearer.

2. The actual survey questions used could not be viewed, but the description in the methods section suggests it to be quite different from the citation. The original was a list of 40 procedures that students rated in their level of confidence using a 5-point scale. The described items also seem to be quite different from this survey, thus calling it a modification may not be most appropriate. With the extensive modification, pre-testing the validity of this survey may be required.  The survey seems to be more focused on endodontic and restorative procedures. If the modifications were in that direction, then perhaps this can be reflected in the title and aims.

3. The authors have classified the study as a prospective study using cohorts of students in the final semester from the years 2018 to 2022, recruiting them in Oct-Dec 2022. Could the authors clarify and address the following:

a.       Have the students from 2018-2021 would have completed the course and graduated? If so there may be intrinsic biases in their response regarding the level of confidence etc. These biases may distort the findngs and should be at least discussed as a limitation.

b.       The study design described seems more of a cross-sectional rather than a prospective design.

4. Could the authors clarify the statement “The questionare showed good internal consistency, with a Cronbach's Alpha score of a=0.83.” in line 153? Were a subset of the students re-surveyed to determine the consistency of responses? If so, this should be stated in the methods section.

5. The statistical analyses should be reported under methods instead of the results. In this case, the T-test may not be suitable for some of the survey questions (eg, parts 2 to 5), where the students rated their self-assessment of their proficiency using 3-point and 5-point scales. They would technically be ordinal data and should not be analysed using the T-test. Moreover, in the tables, it would be more meaningful to report the proportion of subjects each score instead of the F and T statistics.

I look forward to your reply and revision of the manscript.

Comments on the Quality of English Language

In addition, there are several typos with inappropriate hyphens (eg: line 86 “Consequently, individu-als with particular personality profiles may graduate from”

Author Response

The authors have conducted an interesting study evaluating the impact of the COVID-19 pandemic on dental education, specifically in the domains of endodontics and restorative dentistry. This is indeed an important topic in medical and dental education.

-  Dear Editor and reviewers,

Thank you for the revision of our submitted manuscript. The authors feel that all the remarks made by reviewers are well-intended and valuable. We did our best to clarify the issues raised by the reviewers and to improve the quality of our manuscript.

I have several concerns which require clarification:

  1. 1. The details regarding the coursework in the University should come in later after the context and topic have been explained. The introduction should be restructure so that the context is clearer.

- Thank you for this observation, we restructured introduction.

  1. 2. The actual survey questions used could not be viewed, but the description in the methods section suggests it to be quite different from the citation. The original was a list of 40 procedures that students rated in their level of confidence using a 5-point scale. The described items also seem to be quite different from this survey, thus calling it a modification may not be most appropriate. With the extensive modification, pre-testing the validity of this survey may be required. The survey seems to be more focused on endodontic and restorative procedures. If the modifications were in that direction, then perhaps this can be reflected in the title and aims.

- Thank You for sugesstion.

In our Questionnare, we reduced the number of items, which are basically similar the original by Ilic.

  1. The authors have classified the study as a prospective study using cohorts of students in the final semester from the years 2018 to 2022, recruiting them in Oct-Dec 2022. Could the authors clarify and address the following:

  1. Have the students from 2018-2021 would have completed the course and graduated? If so there may be intrinsic biases in their response regarding the level of confidence etc. These biases may distort the findngs and should be at least discussed as a limitation.

-  Thank you for this observation, we added explanatiom in the text.

Yes, they graduated.

The fact that the lack of confidence in the Covid group relative to the anesthetic and endodontic procedures appears due to their inability to do any of these procedures during the Covid period, we organized summer practical school for them under supervision of teachers, and they gain necessary knowledge and self-confidence for these procedures.

  1. The study design described seems more of a cross-sectional rather than a prospective design.

- Thank you for this observation, we corrected in the text.

  1. Could the authors clarify the statement “The questionare showed good internal consistency, with a Cronbach's Alpha score of a=0.83.” in line 153? Were a subset of the students resurveyed to determine the consistency of responses? If so, this should be stated in the methods section.

-Thank You for this observation, we corrected it.

  1.  The statistical analyses should be reported under methods instead of the results. In this case, the T-test may not be suitable for some of the survey questions (eg, parts 2 to 5), where the students rated their self-assessment of their proficiency using 3-point and 5-point scales. They would technically be ordinal data and should not be analysed using the T-test. Moreover, in the tables, it would be more meaningful to report the proportion of subjects each score instead of the F and T statistics.

-Thank You for this observation, we corrected in the article.

- Differences between groups were examined using both non-parametric and parametric statistical methods (difference tests). Since the results of the t-test and the Mann-Whitney U test were without difference, it was decided to present the results using the method of parametric statistics. In support of the choice of the parametric method is also the fact that data follow a normal distribution (not skewed), which was previously checked, and we have an adequate sample size.

  1. In addition, there are several typos with inappropriate hyphens (eg: line 86 “Consequently, individu-als with particular personality profiles may graduate from”

- Thank you for this observation, we corrected inappropriate hyphens in the text.

Round 2

Reviewer 2 Report

Comments and Suggestions for Authors

The authors have not adequately addressed all the points raised in the last revision.

Regarding the author’s reply to my comment: “In our Questionnare, we reduced the number of items, which are basically similar the original by Ilic.”

The questions in this survey significantly differ from the original. Part 1 of the survey in this study focuses on the number of procedures completed and does not tally with the original. In part 2, the use of antibiotics is not discussed in the original survey. Part 6 does not tally with the original survey at all. There are substantial inconsistencies and the above are not exhaustive. This is not just a reduction in the number of questions. It is almost a “new” survey. The original survey is attached in my report and available online (https://journals.plos.org/plosone/article?id=10.1371/journal.pone.0257359)

Regarding the response to “Have the students from 2018-2021 would have completed the course and graduated? If so there may be intrinsic biases in their response regarding the level of confidence etc. These biases may distort the findings and should be at least discussed as a limitation.” The authors’ responses and edits do not capture the biases of surveying a student who has graduated and been in practice. Their assessment of their original competency/confidence etc at the time of graduation will have intrinsic biases that cannot be ignored.

Comments on the Quality of English Language

Author Response

The authors have not adequately addressed all the points raised in the last revision.

Regarding the author’s reply to my comment: “In our Questionnare, we reduced the number of items, which are basically similar the original by Ilic.”

The questions in this survey significantly differ from the original. Part 1 of the survey in this study focuses on the number of procedures completed and does not tally with the original. In part 2, the use of antibiotics is not discussed in the original survey. Part 6 does not tally with the original survey at all. There are substantial inconsistencies and the above are not exhaustive. This is not just a reduction in the number of questions. It is almost a “new” survey. The original survey is attached in my report and available online (https://journals.plos.org/plosone/article?id=10.1371/journal.pone.0257359)

  • Thank You for Your comments. The authors feel that all the remarks are well-intended and valuable. We did our best to clarify the issues and to improve the quality of manuscript.
  • Thank You for observation. This survey are focused on endodontic and restorative procedures. According to Your suggestions, we added in the title that study was focused on endodontic and conservative procedures.
  • Before the research, we perform a pilot study. A small number of dental students participated in the pilot study and results were checked by teachers at the Department of Dental medicine. After the pilot study, we gained a clearer insight into the questionnaire, and for the purposes of the research, we have been modified certain questions related to appropriate procedures in conservative dentistry and Endodontics.
  • Also, suggestions for future research could be dissemination of Questionnaire on large study group, and additional confirmation of validity and reliability.

Regarding the response to “Have the students from 2018-2021 would have completed the course and graduated? If so there may be intrinsic biases in their response regarding the level of confidence etc. These biases may distort the findings and should be at least discussed as a limitation.” The authors’ responses and edits do not capture the biases of surveying a student who has graduated and been in practice. Their assessment of their original competency/confidence etc at the time of graduation will have intrinsic biases that cannot be ignored.

  • Thank You for Your observation.
  • Also, as a limitation of the study, we should consider the fact that the students who participated in the study could show a tendency towards a better self-assessment of knowledge and skills because they are at the final age of their studies, which in their opinion could potentially affect the success and speed of completing their studies. Accordingly, this particular limitation, which concerns the assessment of their original competence/confidence etc at the time of graduation, will have intrinsic biases that cannot be ignored and is highlighted in the framework of potential limitations.